# Comparison of Superselective Renal Artery Embolization versus Retroperitoneal Laparoscopic Partial Nephrectomy in Ruptured Hemorrhagic Renal Angiomyolipoma: A Single-Center Study

**DOI:** 10.3390/diseases12090218

**Published:** 2024-09-16

**Authors:** Zhaoyang Li, Lu Yang, Huitang Yang, Tonghe Zhang, Yandong Cai, Zhan Jiang, Guoju Fan, Kaiqiang Wang, Bo Chen, Hongwei Zhang, Hailong Hu, Yankui Li

**Affiliations:** 1Department of Vascular Surgery, The Second Hospital of Tianjin Medical University, Tianjin 300211, China; lizhaoyang2191@163.com (Z.L.); yanglu880616@163.com (L.Y.); yanghuitang520@126.com (H.Y.); zthzeen@163.com (T.Z.); cyd2020med@163.com (Y.C.); okjcc@163.com (Z.J.); kevinbelin@sina.com (G.F.); wangkaiqiang1117@foxmail.com (K.W.); chenbo01234@126.com (B.C.); zhwwjl@126.com (H.Z.); 2Center for Cardiovascular Diseases, The Second Hospital of Tianjin Medical University, Tianjin 300211, China; 3Department of Urology, The Second Hospital of Tianjin Medical University, Tianjin 300211, China; huhailong@tmu.edu.cn

**Keywords:** renal angiomyolipoma, selective arterial embolization, retroperitoneal laparoscopic part nephrectomy

## Abstract

Purpose: To analyze the clinical efficacy of superselective renal artery embolization and retroperitoneal laparoscopic partial nephrectomy for the treatment of ruptured hemorrhagic renal angiomyolipoma and to provide a reference for the selection of treatment methods for ruptured hemorrhagic renal angiomyolipoma. Methods: A retrospective analysis was conducted on the clinical data of 24 patients who were diagnosed with ruptured hemorrhagic renal angiomyolipoma at the Second Hospital of Tianjin Medical University between January 2019 and December 2021. Among them, 10 patients were treated with superselective arterial embolization (SAE), and 14 patients were treated with retroperitoneal laparoscopic part nephrectomy (RLPN). The differences between the two treatment methods in terms of hospital stay, hospital costs, anesthesia method, operation time, intraoperative blood loss, postoperative bed rest time, antibiotic dosage, postoperative complication rate, tumor diameter changes, creatinine value changes, hemoglobin value changes, tumor recurrence rate, and reoperation rate were compared. Results: All patients completed the treatment and were discharged. There were no significant differences in length of hospital stay, hospital costs, creatinine change values, or postoperative complication rates between the two groups (*p* > 0.05). However, there were statistically significant differences (*p* < 0.05) in surgical time (85.50 ± 19.94 min vs. 141.07 ± 76.33 min), intraoperative blood loss (21.50 ± 14.72 mL vs. 153.57 ± 97.00 mL), postoperative bed rest time (22.7 ± 1.56 h vs. 41.21 ± 3.57 h), preoperative hemoglobin levels (94.7 ± 23.62 g/L vs. 113.79 ± 17.83 g/L), and hemoglobin changes (−6.60 ± 10.36 g/L vs. −15.21 ± 8.79 g/L) between the two groups. Both groups of patients had an average follow-up period of 22 months, and patients in the SAE group had a mean reduction of 3.33 cm in tumor diameter within the follow-up period compared with the pre-embolization period (*p* < 0.05). None of the patients in the SAE group experienced rebleeding, and there was no tumor recurrence in either group. Conclusion: SAE and RLPN are effective treatments for ruptured renal angiomyolipoma with good outcomes. Furthermore, compared to RLPN, SAE offers advantages such as simplicity of operation, minimal trauma, shorter surgical time, minimal impact on hemoglobin levels, shorter bed rest time, faster postoperative recovery, and maximal preservation of renal units.

## 1. Introduction

Renal angiomyolipoma (RAML) is a benign tumor with an incidence of approximately 1/5000–1/15,000 and a maximum age of onset of 40–60 years. The female-to-male ratio of incidence is 2:1 [1]. RAMLs are composed of varying amounts of fat, blood vessels, and smooth muscle [2]. Symptoms of RAML are often nonspecific, with common clinical manifestations including loin pain, hematuria, abdominal masses, and hypotensive shock caused by the rupture of large angiomyolipomas [3]. For patients with acute hemorrhage, partial nephrectomy or nephrectomy was used in the past. With continuous advancements in interventional embolization techniques, more patients with ruptured AML are being treated with selective arterial embolization (SAE) to maximize the preservation of renal function. This study retrospectively analyzed clinical data from the Second Hospital of Tianjin Medical University between January 2019 and December 2021 on treating ruptured RAML using SAE and retroperitoneal laparoscopic part nephrectomy (RLPN) to compare the clinical efficacy of both treatment modalities and provide further guidance for their clinical application.

## 2. Materials and Methods

The clinical data of patients with ruptured hemorrhagic renal angiomyolipomas admitted to the Department of Vascular Surgery and Urology at the Second Hospital of Tianjin Medical University from January 2019 to December 2021 were collected. The study was approved by the Second Hospital of Tianjin Medical University IRB. Patients were divided into SAE and RLPN groups based on the treatment method. General information included sex, age, body mass index (BMI), clinical manifestations (lumbar back pain, soreness, hematuria, nausea, and shock), maximum tumor diameter, multiplicity (single or multiple/bilateral), tumor blood supply (rich blood supply or poor blood supply), and tumor growth pattern (completely endogenous, <50% exogenous growth, or ≥50% exogenous growth).

### 2.1. Inclusion Criteria

Preliminary diagnosis of ruptured hemorrhagic renal angiomyolipoma (RAML) by abdominal color Doppler ultrasound, abdominal CT, or MRI;Patients who presented with abdominal pain, lumbar backache, lumbar back pain, nausea, gross hematuria, or hypovolemic shock (blood pressure < 90/60 mmHg, heart rate > 100 beats/min);Patients who could tolerate selective arterial embolization (SAE) and retroperitoneal laparoscopic partial nephrectomy (RLPN);Availability of complete clinical data.

### 2.2. Exclusion Criteria

Postoperative pathological results confirming nonrenal angiomyolipoma;Patients with concomitant malignant tumors or other renal diseases;Patients who were in poor general condition, had multiple underlying diseases, and were unable to tolerate surgery;Severe coagulation dysfunction and inability to tolerate surgery;Patients with renal artery malformations unsuitable for interventional treatment;Severe contrast agent allergies;Patients and family members who refused surgery and requested conservative treatment;Deceased patients after treatment;Patients with incomplete medical records.

According to the inclusion and exclusion criteria, a total of 24 patients with ruptured hemorrhagic renal angiomyolipoma were included in the SAE and RLPN groups from January 2019 to December 2021 (Table 1 and Table 2).

### 2.3. Definitions

Hemorrhage caused by renal angiomyolipoma rupture included intratumoral hemorrhage, rupture hemorrhage to the renal collecting system, and extrarenal rupture hemorrhage. Depending on the amount of bleeding, it often presents as abdominal distension or discomfort in the waist area, gross hematuria, or hypovolemic shock. Typical radiological manifestations include subcapsular or perirenal hematoma. For critically ill patients, emergency surgical intervention is needed.Maximum tumor diameter: All patients underwent abdominal CT measurements of tumor size before selective arterial embolization (SAE), and the longest diameter of the tumor was measured by imaging.Tumor growth patterns included completely endogenous growth, <50% exogenous growth, and ≥50% exogenous growth [4].Recurrence: Based on radiological and clinical follow-up, a tumor diameter increase >2 cm or the appearance of clinical symptoms requiring further treatment [5].

## 3. Treatment

### 3.1. SAE Treatment

Patients were placed in the supine position and routinely disinfected and toweled. The right common femoral artery was used as the puncture point for all patients. Local anesthesia with 1% lidocaine was administered, and the modified Seldinger method was used for retrograde puncture of the right femoral artery. After successful puncture, a 6F sheath (Cordis, Hialeah, FL, USA) was inserted under the guidance of a 0.035 inch × 150 cm guidewire (Boston Scientific, Alpharetta, GA, USA), and a 5F PIG (Cordis, USA) catheter was advanced to the upper segment of the abdominal aorta. High-pressure injection angiography was performed to locate the opening of the renal artery and determine the number, morphology, and lesion location of the renal arteries, followed by selective renal arteriography. Under the guidance of a 0.035 inch × 150 cm supersmooth guidewire, a 5F C2 catheter (Cordis, USA) was advanced to the opening of the diseased side renal artery. Renal artery openings were localized using high-pressure syringe angiography to determine the number, morphology, and location of the renal arteries. Depending on the size of the target vessel diameter, microguidewires (Asahi Intecc Co., Ltd., Seto-shi, Japan) were used to guide the insertion of either a 1.98Fr or 2.6Fr coaxial microcatheter (Asahi Intecc Co., Ltd., Japan) into distal branches at levels two or three in the renal artery. High-pressure injection angiography again revealed signs such as direct bleeding from the extravasation contrast agent and tortuous enlargement with structural disorder along with vascular composition outlining tumor contours in blood supply arteries feeding into tumors. After careful assessment of the tumor’s blood supply, a suspension of polyvinyl alcohol (PVA) embolic agent (COOK, Bloomington, IN, USA) mixed with an appropriate amount of contrast agent (usually at a concentration of 30% to 40%) was prepared. Under vascular angiography and fluoroscopy, the PVA particle suspension was injected into the tumor-feeding artery through a microcatheter at an appropriate speed to avoid reflux. For patients with concomitant arterial aneurysms, embolization coils (COOK, USA) of suitable diameter and quantity were used for embolization based on the target vessel diameter. For patients without arterial aneurysms, if incomplete embolization was observed after the use of PVA particles, spring coils were used to embolize the main tumor-feeding artery. In cases where the tumor was supplied by multiple arteries, selective embolization of each branch vessel with appropriately sized PVA was performed to minimize damage to normal renal units. Subsequent DSA revealed no distal visualization or extravasation of contrast agent in the branch vessels supplying the renal tumors. The right femoral artery puncture site was closed using a vascular closure device followed by hemostatic compression and pressure bandaging. After completion of the procedure, the patient returned to the ward for bed rest with immobilization of the right lower limb for 24 h while being closely monitored for any signs of bleeding from the puncture site (Figure 1).

### 3.2. RLPN Treatment

After successful tracheal intubation under general anesthesia, patients were placed in the healthy lateral position with the waist bridge elevated. The skin of the surgical area was routinely disinfected, and drapes were laid out. A horizontal incision approximately 3.0 cm long was made below the 12th rib on the posterior axillary line of the affected side, followed by blunt dissection to the retroperitoneal space. The peritoneum was pushed inward with the fingers, and a disposable expansion balloon was inserted, into which 600 mL of air was injected to expand the retroperitoneum. After approximately 5 min, the balloon was removed. Under finger guidance, a 12 mm trocar was inserted and secured at two points: below the 12th rib on the posterior axillary line of the affected side and 2 cm above the iliac crest on the midaxillary line. Then, a laparoscope was introduced into the abdominal cavity through these trocars. A trocar with a diameter of 5 mm was also inserted below the anterior axillary line at the 12th rib under direct vision of a laparoscope, maintaining CO_2_ pneumoperitoneum pressure at 13 cm H_2_O. The extraperitoneal fat tissue was dissected bluntly and sharply under direct vision using a laparoscope along with an ultrasonic knife from the renal hilum toward the renal pedicle to free up the kidney along its fascia plane until reaching the renal hilum pulsating artery for identification and isolation; then, the kidney was freed from its lower pole while fully exposing the tumor after defining the boundary between the tumor mass and normal renal parenchyma. Subsequently, nondamaging vascular clamps were used to occlude the renal artery. At the edge of the swollen area, at a distance of approximately 0.5 cm, the blunt and sharp combination was used to separate the tumor from the renal parenchyma, and 2–0 absorbable barbed sutures were used to suture the rupture of the collecting system and renal parenchymal blood vessel ends. Then, a size 0 absorbable barbed suture was used to suture the renal parenchymal section in two layers without leaving any residual cavity between them. After confirming that there was no bleeding at the incision site, the vascular clamp was removed; if no significant active bleeding in the renal parenchyma was observed, hemostatic gauze was placed there and an incision was made below the 12th rib on the posterior axillary line to remove the kidney mass to stop the bleeding thoroughly. The wound was rinsed, and no active bleeding was checked for. A drainage tube was placed in the retroperitoneal cavity after checking the instruments and dressings. The incision was closed layer by layer, and the operation was completed after the endoscope and working sheath were removed.

## 4. Observation Indicators

The two treatment methods were compared in terms of length of hospital stay, hospitalization costs, surgical time, intraoperative blood loss, incision length, renal artery occlusion time, drainage tube removal time, postoperative bed rest time, creatinine change value, hemoglobin change value, postoperative complication rate, tumor recurrence within an average follow-up period of 22 months after surgery, and tumor diameter reduction in the SAE group.

## 5. Statistical Methods

The original clinical data obtained in this study were screened, processed, and analyzed using SPSS 21.0 statistical analysis software. The frequency data are expressed as percentages (%), and the chi-square test was used for comparisons. Continuous data are presented as means ± standard deviations (SDs), and an independent-sample *t*-test was used for pairwise comparisons. The rank-sum test was used for ordinal data. A significance level of *p* < 0.05 was defined as a statistically significant difference.

## 6. Results

The SAE group consisted of 10 patients, with an average hospital stay of 11.80 ± 4.75 days and an average hospital cost of USD 5649.21 ± 1742.09. The mean surgical time was 85.50 ± 19.94 min, with an average intraoperative blood loss of 21.50 ± 14.72 mL and a postoperative bed rest time of 22.7 ± 1.56 h. The postoperative creatinine level averaged 62.57 ± 21.57 μmol/L, while the postoperative hemoglobin level averaged 94.70 ± 23.62 g/L. The average change in creatinine was 5.15 ± 12.01 μmol/L, and the average change in hemoglobin level was 6.60 ± 10.36 g/L. There were four cases of postoperative complications, including one case of liver function abnormality, which improved after hepatoprotective treatment; two cases of fever accompanied by leukocytosis, possibly due to the absorption of necrotic material after embolization, which improved after anti-inflammatory and antipyretic treatment; and one case of renal area pain, possibly due to local ischemia after embolization, which improved after analgesic treatment.

The RLPN group consisted of 14 patients, with an average hospital stay of 9.50 ± 2.27 days and an average hospital cost of USD 4969.79 ± 512.93. The mean surgical time was 141.07 ± 76.33 min, with an average intraoperative blood loss of 153.57 ± 97.00 mL. The average postoperative bed rest time was 41.21 ± 3.57 h, while the postoperative creatinine level averaged 86.15 ± 49.82 μmol/L and the postoperative hemoglobin level averaged 113.79 ± 17.83 g/L. The mean change in the creatinine value was 9.59 ± 22.61 μmol/L, and the mean change in the hemoglobin value was 15.21 ± 8.79 g/L. The average incision length measured 3.00 ± 1.17 cm, with a mean renal artery occlusion time of 20.29 ± 6.00 min and a mean drainage tube removal time of 5.14 ± 1.46 days. All patients received local hemostasis with hemostatic gauze during surgery. Nine patients experienced complications postoperatively, including fever accompanied by leukocytosis (two patients), incisional pain (four patients), nausea/vomiting (two patients), and cough/sputum production (one patient). All complications improved after symptomatic treatment.

In the surgical data of the patient group, there were no statistically significant differences in hospital stay, hospital costs, or creatinine changes (*p* > 0.05). However, there were statistically significant differences in surgical time, intraoperative blood loss, hemoglobin changes, and postoperative bed rest time (*p* < 0.05). The SAE group underwent interventional surgery with only puncture sites needed, without general anesthesia or renal artery blockade or drainage tube placement (Table 3, Figure 2).

## 7. Follow-Up

The average follow-up period after surgery for both groups of patients was 22 months, and all patients received outpatient follow-up. In the SAE group, postembolization re-examination via bilateral renal CT revealed a significant reduction in tumor diameter, indicating remarkable therapeutic efficacy. During the follow-up period, neither group underwent further surgery, and there was no tumor recurrence. In the SAE group, the average tumor diameter decreased by 3.33 cm compared to that in the pre-embolization group (*p* < 0.05) (Table 4).

## 8. Discussion

The majority of RAML patients are asymptomatic when they are detected during physical examination. The main manifestation of RAML with ruptured bleeding is Wunderlich Syndrome [6], which may be related to internal vascular malformations in the tumor, secondary arterial aneurysm formation, and a large tumor diameter [7]. In recent years, research has shown that a tumor diameter > 4 cm is not the only determining factor for spontaneous ruptured bleeding in RAML; it is also possible for spontaneous ruptured bleeding to occur with a diameter < 4 cm [5]. Tumor rupture and bleeding are also associated with genetic factors, a rich tumor blood supply, pregnancy, estrogen levels, coagulation disorders, trauma, and tuberous sclerosis [8,9,10]. RAML patients can be treated with observation, medication [11,12], and surgery. Surgical options for RAML patients include tumor excision, partial nephrectomy, radical nephrectomy surgery (RNS), and SAE. In an emergency, RNS can promptly save the patient’s life. Nephron-sparing surgery (NSS) as a treatment for preserving renal units may lead to postoperative tumor recurrence and common complications such as urinary leakage, fistula formation, bleeding, and intestinal obstruction [13,14,15]. Before implementing treatment, clinical physicians must fully explain the condition to patients and their families and the pros and cons of various treatment methods to make the best choice based on a thorough understanding of the condition.

This study conducted a comparative analysis of the surgical and follow-up data of patients in the SAE and RLPN groups. There was no statistically significant difference (*p* > 0.05) in terms of length of hospital stay, hospital costs, postoperative creatinine change, or incidence of complications. However, there were statistically significant differences (*p* < 0.05) regarding the incision length, intraoperative blood loss, surgical time, and hemoglobin change. The SAE group was superior to the RLPN group regarding anesthesia method, intraoperative blood loss, surgical time, postoperative bed rest time, and hemoglobin change. In addition to these findings, the SAE procedure is minimally invasive, with only a puncture site required without the need for renal artery occlusion or drainage tube placement. One patient in the SAE group received a transfusion of 8 units of packed red blood cells due to excessive blood loss; however, no patients required transfusion in the RLPN group. Each patient received local application of hemostatic gauze at the wound site, which may have influenced the assessment of changes in hemoglobin.

When ruptured and bleeding RAMLs have a large tumor diameter, multiple bilateral occurrences, and deep locations, especially the exophytic type, the tumor ruptures and bleeds outside the kidney, forming a perirenal or retroperitoneal hematoma [16]. The poor visual field of retroperitoneal laparoscopic surgery and limited surgical space lead to increased surgical difficulty, prolonged operation time, and increased intraoperative blood loss. In this study, one patient in the SAE group had a maximum tumor diameter of 14.7 cm, a surgery time of 98 min, and an intraoperative blood loss of 35 mL. The tumor in this patient was solitary, and SAE surgery was simple. The difficulty of surgery is related to whether there is renal artery malformation or not, as well as the number of arteries supplying the tumor; it is not significantly related to the multiplicity, side involvement, growth pattern, or rupture status of the tumor. One patient in the RLPN group had a maximum tumor diameter of 14.1 cm with a surgery time of 260 min and an intraoperative blood loss amounting to 150 mL. In comparison, SAE treatment has advantages.

SAE can lead to ischemic necrosis of tumor cells, a significant reduction in vascular components, and smooth muscle atrophy, thereby slowing the growth rate of tumors and minimizing damage to normal renal tissue [17,18]. In emergency patients with acute rupture bleeding, rebleeding after embolization, or unstable hemodynamics, timely embolization can not only save the patient’s life but also preserve normal functioning renal units [19]. NSS combined with SAE can significantly reduce the difficulty of tumor and kidney removal and the incidence of postoperative complications [20,21,22,23]. In this study, SAE and RLPN showed significant efficacy in treating ruptured AML. Compared with RLPN, SAE treatment is less traumatic and is associated with shorter hospital stays, rapid hemostasis, maximum preservation of renal units, and life-saving benefits. It is particularly suitable for patients with acute bleeding AML [18].

Compared with RLPN, SAE cannot completely remove the tumor mass and cannot determine the nature of the tumor. If the tumor does not shrink well, recurs, or bleeds again, a second embolization may be required [24,25,26]. In addition, common complications of SAE include postembolization syndrome, vascular injury, renal infarction with abscess formation, and nontarget embolization. Postembolization syndrome is the most common complication and is characterized mainly by fever, abdominal pain, nausea, vomiting, and leukocytosis. These symptoms are mostly self-limiting and can also be improved through conservative treatments, such as antipyretics, antiemetics, and analgesics [20,24,27,28,29]. In this study, four patients were treated for SAE after surgery: one patient with abnormal liver function (Clavien-Dindo I), two patients with fever accompanied by leukocytosis (Clavien-Dindo I), and one patient with renal area pain (Clavien-Dindo I). However, no severe complications, such as nontarget lesion embolism, renal abscess, or renal infarction, occurred in this group, whereas there were nine cases of postoperative complications in the RLPN group, two cases of fever accompanied by leukocytosis (Clavien-Dindo I), four cases of incisional pain (Clavien-Dindo I), two cases of nausea and vomiting (Clavien-Dindo I), and one case of sputum and cough (Clavien-Dindo I). Severe complications, such as renal insufficiency, urinary leakage, fistula formation, or pulmonary infection, did not occur in this group either.

During the follow-up period, the average diameter reduction was 3.33 cm in the SAE group. No tumor recurrence, rebleeding, or secondary surgery was observed in either group. In conclusion, SAE and RLPN are effective in treating ruptured hemorrhagic renal angiomyolipoma. SAE is an efficient and effective technique for the emergency treatment of ruptured AML, significantly reducing tumor size and lowering recurrence rates. The safety of embolization after the procedure is also satisfactory [30].

## 9. Conclusions

SAE and RLPN are effective in treating ruptured hemorrhagic renal angiomyolipoma. For patients with ruptured hemorrhagic renal angiomyolipoma, bilateral/multiple lesions, large diameter, deep location (near the renal hilum, compressing the renal pelvis), rich blood supply, high surgical risk (severe cases may require nephrectomy), and critical illness requiring rescue treatment, superselective arterial embolization has advantages over retroperitoneal laparoscopic partial nephrectomy in terms of simple operation, minimal trauma, short surgical time, low intraoperative blood loss, minimal impact on hemoglobin levels, shorter bed rest time, faster postoperative recovery, and maximum preservation of renal units.

## Figures and Tables

**Figure 1 diseases-12-00218-f001:**
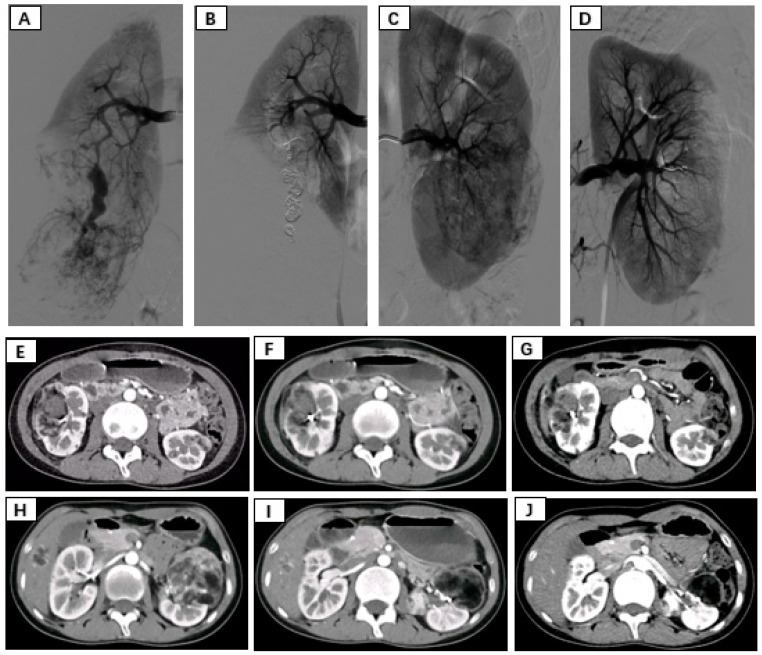
(**A**,**B**) Immediate pre- and postembolization imaging of the right renal angiomyolipoma. (**C**,**D**) Immediate pre- and postembolization imaging of the left renal angiomyolipoma. (**E**–**G**) Rupture and hemorrhage of the right renal angiomyolipoma, followed by 3-month, 6-month, and 12-month follow-up contrast-enhanced CT showing a significant reduction in tumor size. (**H**–**J**) Pre-embolization, 3-month postembolization, and 6-month postembolization follow-up contrast-enhanced CT images of the left renal angiomyolipoma demonstrating a marked reduction in tumor size.

**Figure 2 diseases-12-00218-f002:**
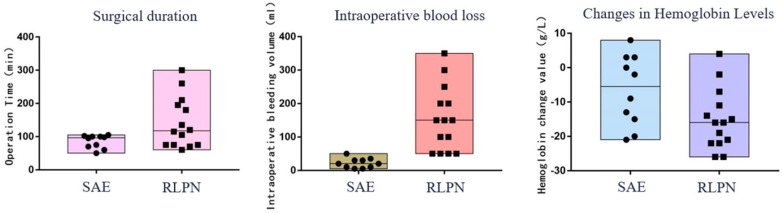
There were statistically significant differences in surgical time, intraoperative blood loss, and changes in hemoglobin between the two groups (*p* < 0.05).

**Table 1 diseases-12-00218-t001:** Comparison of the general data for the SAE group and the RLPN group.

	SAE Group (*n* = 10)	RLPN Group (*n* = 14)	*p* Value
Age (years)	44.00 ± 15.51	54.29 ± 11.83	0.078
Gender			0.079
Female	9 (90%)	7 (50%)	
Male	1 (10%)	7 (50)	
BMI (kg/m^2^)	26.12 ± 4.43	25.79 ± 2.93	0.827
Diameter on CT (cm)	9.53 ± 3.33	5.60 ± 2.94	0.008
<4	1 (10%).	4 (28.57%)	
≥4	9 (90%)	10 (71.43%)	
Multiplicity			0.010
Single	3 (30%)	12 (85.71%)	
Multiple	7 (70%)	2 (14.29%)	
Tumor growth pattern			0.193
Fully endogenous	3 (30%)	1 (7.14%)	
<50% of exogenous growth	2 (20%)	7 (50%)	
≥50% exogenous growth	5 (50%)	6 (42.86%)	
Tumor blood supply			0.615
Hypervascularity	9 (90%)	11 (78.57%)	
Lack of blood supply	1 (1/10)	3 (21.43%)	
Creatinine (mu mol/L)	57.42 ± 17.02	76.56 ± 33.28	0.068
Hb (g/L) (1 case of infusion of SAE Group 8 units suspended red blood cells, did not receive further blood transfusion)	101.30 ± 21.38	129.00 ± 20.48	0.005

**Table 2 diseases-12-00218-t002:** Comparison of clinical manifestations between the SAE group and the RLPN group.

Clinical Signs and Symptoms	SAE Group	RLPN Group	*p* Value
Abdominal pain	7	2	0.010
Lower back pain	7	6	0.240
Sore waist and back	3	6	0.678
Hematuria	1	2	1.000
Nausea	2	0	0.163
Shock	3	1	0.272

**Table 3 diseases-12-00218-t003:** Comparison of surgical data between SAE and RLPN groups.

	SAE Group	RLPN Group	*p* Value
Hospital stay (days)	11.80 ± 4.75	9.50 ± 2.27	0.128
Hospital costs (USD)	41,055.27 ± 12,660.51	36,117.65 ± 3727.67	0.179
Surgical time (min)	85.50 ± 19.94	141.07 ± 76.33	0.020
Surgical incision length (cm)	-	3.00 ± 1.17	0.000
Postoperative bed rest time (h)	22.7 ± 1.56	41.21 ± 3.57	0.000
Creatinine changes (μmol/L)	5.15 ± 12.01	12.92 ± 20.85	0.303
Hemoglobin changes (g/L)	−6.60 ± 10.36	−15.21 ± 8.79	0.039
Renal artery blockade time (min)	-	20.29 ± 6.00	0.000
Drainage tube removal time (days)	-	5.14 ± 1.46	0.000
Postoperative complication rate (%)	40.0 (4/10)	64.3 (9/14)	0.408

**Table 4 diseases-12-00218-t004:** Comparison of tumor diameters before and after interventional surgery in the SAE group.

	Preoperative	After the Intervention	Average Reduction Diameter	*p* Value
Tumor diameter (cm)	8.67 ± 3.12	6.22 ± 2.53	3.33 ± 1.12	0.000

## Data Availability

Data are contained within the article.

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
