# Peer review of "Comparison of Superselective Renal Artery Embolization versus Retroperitoneal Laparoscopic Partial Nephrectomy in Ruptured Hemorrhagic Renal Angiomyolipoma: A Single-Center Study"

_diseases, 2024, doi:10.3390/diseases12090218_

Round 1
Reviewer 1 Report
Comments and Suggestions for Authors
Thank you for your work;
Even if is a non randomized trial, the number you presented are relevant for AMLs.
There are some issues you must clarify:
Why you decide for SAE vs LP treatment?
What is the indication for the two treatments?
As emerging for your paper there is limited or no space for conservative managements in potentially stable roptured AML.
Can you add the coagulative status of these patients? ( antiplatelet- anticoagulant etc.)
Any known genetic condition / familiarity in your patient cohort?
To improve terminology please use the Historical name of this Syndrome:
Wunderlich Syndrome(Shah JN, Gandhi D, Prasad SR, Sandhu PK, Banker H, Molina R, Khan S, Garg T, Katabathina VS. Wunderlich Syndrome: Comprehensive Review of Diagnosis and Management. Radiographics. 2023 Jun;43(6):e220172. doi: 10.1148/rg.220172. Erratum in: Radiographics. 2023 Jul;43(7):e239007. doi: 10.1148/rg.239007. PMID: 37227946.)
Retroperitoneal Laproscopy can be very tricky in case of anterior/superior pole position of lesions (more than just dimension or growth pattern), in case of active bleeding like this case serie.
This can lead to unsuccesful conservative management and an higher risk of radical nephrectomy.
Did you use a 4th trocar for suction/irrigation?
How did you estimated blood loss in SAE group?
Conservative management should always be considered for this clinical setting,
SAE should be taken always in account for first management, considering these bleedings as kidney traumas.
Probably the only setting in wich surgery has a preminent role is the haemodinamyc instability or the inavalaibility of interventional radiology.
Add in discussion references about the bleeding risk among dimensional categories of AML ( multifocality/ 4 cm)
and just add a mention on the proposed introduction of TKI for AML.
Comments on the Quality of English Language
minor english editing
Author Response
1) Why you decide for SAE vs LP treatment?
Response: Thank you for your suggestion. SAE and RLPN are two effective minimally invasive procedures for the treatment of ruptured hemorrhagic renal angiomyolipoma (AML), but they have not been compared in the treatment of ruptured hemorrhagic renal AML. In this study, we retrospectively investigated the efficacy of these two surgical modalities in the treatment of ruptured hemorrhagic renal AML to provide a reference for guiding the choice of treatment modality.
2) What is the indication for the two treatments?
Response: Thank you for your suggestion. Regardless of the treatment modality, preservation of renal function should be a priority. Tumors >4 cm have an increased risk of rupture and hemorrhage, and surgery to preserve the renal unit may be considered. SAE is the first-line treatment for active hemorrhagic renal AML [1, 2]. Surgery is the treatment of choice when tumor rupture and bleeding is unconditional for SAE, and the choice of surgery should be based on hemostasis and removal of the tumor as much as possible to preserve the normal renal unit. When surgical exploration is used for acute, life-threatening bleeding, the kidney is often removed, and with advances in technology, RLPN is now more often used.
3) Can you add the coagulative status of these patients? ( antiplatelet- anticoagulant etc.)
Response: Thank you for your suggestion. The 24 patients with ruptured hemorrhagic renal angiomyolipoma included in this study were not administered antiplatelet drugs, anticoagulants, etc. The preoperative relevant coagulation indices are listed below.
|
Preoperative relevant coagulation indices |
||||
|
|
|
t |
p |
|
|
SAE |
PN |
|||
|
PLT |
249.08±110.87 |
226.33±50.15 |
0.648 |
0.524 |
|
PT (s) |
11.02±0.91 |
10.30±0.37 |
2.364 |
0.037* |
|
APTT (s) |
26.83±2.55 |
24.53±2.86 |
1.998 |
0.059 |
|
Fbg (g/L) |
3.57±1.32 |
2.67±0.60 |
2.180 |
0.041* |
|
TT (s) |
17.63±1.40 |
18.99±1.69 |
-2.055 |
0.053 |
|
* p<0.05 ** p<0.01. |
||||
4) Any known genetic condition / familiarity in your patient cohort?
Response: Thank you for your suggestion. Renal angiomyolipoma can be a stand-alone disease or a manifestation of tuberous sclerosis (TSC). The relatively short duration of this study does not yet allow for a clear correlation with genetic conditions/familiarity.
5) To improve terminology please use the Historical name of this Syndrome:
Wunderlich Syndrome(Shah JN, Gandhi D, Prasad SR, Sandhu PK, Banker H, Molina R, Khan S, Garg T, Katabathina VS. Wunderlich Syndrome: Comprehensive Review of Diagnosis and Management. Radiographics. 2023 Jun;43(6):e220172. doi: 10.1148/rg.220172. Erratum in: Radiographics. 2023 Jul;43(7):e239007. doi: 10.1148/rg.239007. PMID: 37227946.)
Response: Thank you for your suggestion. We agree with this comment. Reference No.6 is PMID: 37227946.
6)Did you use a 4th trocar for suction/irrigation?
Response: Thank you for your suggestion. Typically, we do not use a 4th trocar for suction/irrigation or an intraoperative drainage tube through the puncture wound; when a patient is obese and difficult to maneuver, a 4th puncture port can be used to assist in treatment.
7) How did you estimated blood loss in SAE group?
Response: Thank you for your suggestion. The average amount of intraoperative bleeding in the SAE group was 21.50±14.72 ml, which we estimated on the basis of the amount of blood contained in the gauze during surgery[3].
8) Add in discussion references about the bleeding risk among dimensional categories of AML ( multifocality/ 4 cm) and just add a mention on the proposed introduction of TKI for AML.
Response: Thank you for pointing this out. We agree with this comment. Therefore, we have added Reference No.8,9,10 concerning the bleeding risk among the dimensional categories of AML and the TKI treatment of AML(Reference No.11) to the Discussion section.
- Murray, T. E.; Lee, M. J., Are We Overtreating Renal Angiomyolipoma: A Review of the Literature and Assessment of Contemporary Management and Follow-Up Strategies. Cardiovasc Intervent Radiol 2018, 41 (4), 525-536.
- Vaggers, S.; Rice, P.; Somani, B. K.; Veeratterapillay, R.; Rai, B. P., Evidence-based protocol-led management of renal angiomyolipoma: A review of literature. Turk J Urol 2021, 47 (Supp. 1), S9-s18.
- Ali Algadiem, E.; Aleisa, A. A.; Alsubaie, H. I.; Buhlaiqah, N. R.; Algadeeb, J. B.; Alsneini, H. A., Blood Loss Estimation Using Gauze Visual Analogue. Trauma Mon 2016, 21 (2), e34131.
Reviewer 2 Report
Comments and Suggestions for Authors
I read with great interest this manuscript. The authors retrospectively analyzed clinical data of their centers concerning treatment of ruptured renal AML, comparing SAE and retroperitoneal laparoscopic partial nephrectomy. Despite the overall lack of novelty I believe that real world contemporary descriptive data are well accepted. However, I would suggest some improvements:
- according to post-op complications report, please adopt Clavien- Dindo score to classify them
- please specify how you performed follow-up of these patients. DId you repeat a CT-scan in the first 30 days after surgery?
- SAE has been also standardised in the field of clampless laparoscopic partial nephrectomy, reducing the potential ischemic damage deriving from clamping the main renal artery during renal tumor dissection . In order to enrich discussion section and improve the overall quality of the manuscript, please discuss such topic and cite PMID: 17445641, PMID: 21797763, PMID: 19694532
Author Response
1) according to post-op complications report, please adopt Clavien- Dindo score to classify them
Response: Thank you for your suggestion. We agree with this comment. Clavien-Dindo scores are highlighted in the text. The table is listed below.
|
Postoperative Clavien‒Dindo classification of RAML |
||
|
Clavien‒Dindo Grade |
SAE |
RLPN |
|
Grade â… |
4 |
9 |
|
Grade â…¡ |
0 |
0 |
|
Grade â…¢ |
0 |
0 |
|
Grade â…£ |
0 |
0 |
|
Grade â…¤ |
0 |
0 |
2) please specify how you performed follow-up of these patients. DId you repeat a CT-scan in the first 30 days after surgery?
Response: Thank you for your suggestion. We referred to the renal tumor follow-up approach, which generally involves 3-, 6-, 12-, and 24-month ultrasound or CT follow-up. CT scans were not repeated in the first 30 days after surgery.
3) SAE has been also standardised in the field of clampless laparoscopic partial nephrectomy, reducing the potential ischemic damage deriving from clamping the main renal artery during renal tumor dissection . In order to enrich discussion section and improve the overall quality of the manuscript, please discuss such topic and cite PMID: 17445641, PMID: 21797763, PMID: 19694532
Response: Thank you for your suggestion. We agree with this comment. Reference No.21 is PMID: 19694532. Reference No.22 is PMID: 21797763.Reference No.23 is PMID: 17445641.
Round 2
Reviewer 1 Report
Comments and Suggestions for Authors
Thank you for adding missing informations
Bleeding in AML arise from kidney
in SAE group you considered blodd in gauze as indicator of bleeding, I think is useless, you can just use Hb dynamics.
Thanks for adding requested bibliographic notes